# A Newly Developed Scale for Assessing Experienced and Anticipated Sexual Stigma in Health-Care Services for Gay and Bisexual Men

**DOI:** 10.3390/ijerph192113877

**Published:** 2022-10-25

**Authors:** Mei-Feng Huang, Yu-Ping Chang, Chung-Ying Lin, Cheng-Fang Yen

**Affiliations:** 1Department of Psychiatry, School of Medicine, College of Medicine, Kaohsiung Medical University, Kaohsiung 80708, Taiwan; 2Department of Psychiatry, Kaohsiung Medical University Hospital, Kaohsiung 80756, Taiwan; 3School of Nursing, The State University of New York, University at Buffalo, Buffalo, NY 14260, USA; 4Institute of Allied Health Sciences, College of Medicine, National Cheng Kung University, Tainan 70101, Taiwan; 5Biostatistics Consulting Center, National Cheng Kung University Hospital, College of Medicine, National Cheng Kung University, Tainan 70101, Taiwan; 6Department of Public Health, College of Medicine, National Cheng Kung University, Tainan 70101, Taiwan; 7Department of Occupational Therapy, College of Medicine, National Cheng Kung University, Tainan 70101, Taiwan; 8College of Professional Studies, National Pingtung University of Science and Technology, Pingtung 91201, Taiwan

**Keywords:** stigma, gay, bisexual, instrument, mental health

## Abstract

Gay and bisexual men’s experiences and worries of sexual stigma by health-care providers may delay them from seeking health-care assistance. Our study developed the Experienced and Anticipated Sexual Stigma Scale in Health-care Services (EASSSiHS) and examined its psychometric properties. The six-item EASSSiHS was first developed on the basis of the results of focus group interviews with 24 participants. Parallel analysis was used to determine the number of factors. Exploratory factor analysis (EFA) and confirmatory factor analysis (CFA) were performed to examine the factor structure. The internal consistency was examined using McDonald’s omega coefficient. Concurrent validity was examined using Pearson correlations with perceived sexual stigma from family and friends, depression, anxiety, and loneliness. The results of parallel analysis and EFA indicated that the factor structure of the EASSSiHS included two factors: experienced stigma and anticipated stigma. The result of CFA further confirmed the two-factor structure. The EASSSiHS had good internal consistency and acceptable concurrent validity. The anticipated stigma factor had stronger associations with perceived sexual stigma from family members and friends, depression, anxiety, and loneliness, when compared with the experienced stigma factor. The results of this study supported the psychometric properties of the EASSSiHS for assessing experienced and anticipated sexual stigma in health-care services among gay and bisexual men. The experience and worry of sexual stigma in health-care services were not rare among gay and bisexual men; therefore, interventions to enhance the cultural competency of health-care workers are urgently required.

## 1. Introduction

Researchers have identified health disparities between lesbian, gay, and bisexual (LGB) individuals and heterosexual populations [1]. For example, studies have indicated elevated rates of several physical diseases, such as human immunodeficiency virus (HIV) infection [2], cancer [3], diabetes [4], and hypertension [5], among LGB individuals. Moreover, the rates of psychiatric disorders, substance use, violence, self-harm, and suicide were higher among LGB individuals than heterosexuals [6,7]. According to minority stress theory [8], these health disparities may be partially due to sexual stigma, including prejudice, stereotypes, and discrimination toward LGB individuals. A systematic review and meta-analysis study demonstrated that LGB individuals are more likely to experience health inequalities due to heteronormativity-related sexually stigmatizing victimization and discrimination [9]. However, sexual stigma toward LGB individuals is also common among health-care providers [10,11,12]. Experiences of mistreatment due to sexual stigma (experienced stigma) promote concealment of sexual orientation and impede health-seeking behaviors among LGB individuals [13,14]. Earnshaw and Chaudoir proposed a framework for illustrating the influence of sexual stigma in health-care services [15] based on the premise that LGB individuals may expect future mistreatment (anticipated stigma) or even endorse sexual stereotypes and prejudice and avoid accessing medical care. In addition to promoting cultural sensitivity training for health-care providers, research must evaluate experienced and anticipated sexual stigma, in conjunction with health-care-seeking among LGB individuals.

Several researchers have developed survey instruments for measuring the mistreatment experienced by LGB individuals during interactions with health-care providers [13,16]; however, limited survey instruments have been developed to assess the experienced and anticipated sexual stigma in health-care services among LGB individuals. Grosso et al. developed a stigma measure to evaluate the experiences of sexual stigma from family members, friends, police officers, strangers, and health-care workers among men who have sex with men (MSM) and female sex workers in Sub-Saharan Africa [17]. However, evaluating multiple sources of sexual stigma simultaneously may increase clients’ bias in identifying the sources of sexual stigma. Evaluating multiple sources of sexual stigma simultaneously may also reduce clients’ incentive to participate in the survey because of increased cognitive and emotional loading. It is necessary to develop a measure to specifically assess LGB individuals’ experiences and worries about sexual stigma when seeking medical care. Asian societies have a lower tolerance of gay and bisexual men than Western societies [18]. Public stigma toward gay and bisexual men is prevalent in Taiwan [19], and sexual stigmatization experienced by gay and bisexual men in health-care services warrants in-depth study. Furthermore, studies have shown that perceived sexual stigma from the public, family members, and peers was significantly associated with mental health problems among gay and bisexual men [19,20]. Whether the experiences and worries of sexual stigma are related to mental health problems among gay and bisexual men requires further study.

We developed the Experienced and Anticipated Sexual Stigma Scale in Health-care Service (EASSSiHS) and examined its psychometric properties. Studies have found that experiencing sexual stigma in health-care services was significantly associated with perceived sexual stigma from the public and close persons [21]. According to minority stress theory [8], experiencing or worrying about sexual stigma in health-care services is a psychological stressor for gay and bisexual men and compromises their mental health. Moreover, there were gender differences in the experienced and anticipated sexual stigma between gay and bisexual men and lesbian and bisexual women [22]. Therefore, we examined the concurrent validity of the EASSSiHS by testing its correlations with perceived sexual stigma from family and friends, depression, anxiety, and loneliness among gay and bisexual men.

## 2. Materials and Methods

### 2.1. Participants and Procedure

The participant inclusion criteria were men who identified as gay or bisexual, aged 20 or older, and living in Taiwan. Participants were recruited by posting an online advertisement on social media, including Facebook, Twitter, and LINE (a direct messaging app), the bulletin board system, from August 2021 to May 2022, starting after the end of the first severe COVID-19 outbreak (between May and July 2021) and ending at the beginning of the second severe outbreak (since May 2022 to now). We also posted advertisements on the home pages of three health promotion and counseling centers for LGB individuals. The three centers were organized by gay- and lesbian-friendly groups; advertisement on their home pages could gain attention and trust from gay and bisexual men. Interested potential participants were asked to telephonically contact the study’s research assistants, who ensured the eligibility of potential participants, explained the study aims and procedures, and scheduled a time for eligible participants to complete the study questionnaires individually in a quiet study room. The research assistants evaluated the participants in the on-site study room to determine whether they had impaired intellect or showed signs of alcohol and substance use that might interfere with their understanding of the study’s purpose or completing the questionnaire. In total, 736 gay or bisexual men participated in the study. No participant was excluded. Informed consent was obtained from all participants before the assessment. The study was approved by the Institutional Review Board of Kaohsiung Medical University Hospital (KMUHIRB-F(I)-20210003).

### 2.2. Measures

#### 2.2.1. Development Process of the EASSSiHS

Before beginning the formal research, we conducted three focus group interviews from March to May 2021 to help develop the EASSSiHS assessing the experienced and anticipated sexual stigma when receiving health-care services among gay and bisexual men. We recruited the focus group participants by posting an online advertisement on the home pages of three health promotion and counseling centers for sexual minorities. The recruitment criteria were gay or bisexual men aged 20 or older living in Taiwan. A total of 24 gay or bisexual men participated in focus group interviews, with 8 participants in each group. The participants’ mean age was 30.1 years (standard deviation [SD] = 3.4), 91.7% of participants identified as gay and 8.3% as bisexual, and 87.5% of participants had completed college or university studies. According to the results of literature review and the aims of this study, we determined the discussion topics for the focus groups as the experiences and worries of receiving treatment that differed from their expectations and different from that received by others due to the health-care providers’ awareness or suspicion of participants’ sexual orientation. The principal investigator led the group discussion and encouraged the members to express their opinions. Three researchers reviewed the transcript and coded the data for indications of sexual stigma experienced or anticipated by gay and bisexual men. The principal investigator reviewed the coding results and integrated them into six items of the EASSSiHS, including being rejected from receiving health-care services, receiving inferior health-care services, being gossiped about because of one’s sexual orientation, having difficulties in obtaining health-care services, being afraid of seeking health-care services, and avoiding seeking health-care services. Each item was answered with “yes” or “no”.

#### 2.2.2. Homosexuality-Related Stigma

Homosexual-related stigma was measured using the homosexuality-related stigma scale (HRSS). The HRSS used in the present study contains 12 items with responses ranging from *strongly disagree* (score: 1) to *strongly agree* (score: 4). A summed HRSS score was used in the present study, and a higher HRSS score indicated higher levels of stigma toward homosexuality that the participants perceived from their family members [23]. The HRSS has been found to be a valid and psychometrically sound instrument, including its Taiwan version [24,25]. The present study’s Cronbach’s alpha coefficient (α) was 0.93.

#### 2.2.3. Depression

Depression among the participants was measured using the Center for Epidemiologic Studies Depression Scale (CES-D). The CES-D contains 20 items with responses ranging from *rarely or none of the time* (*less than 1 day*) (score: 0) to *most or all of the time* (*5–7 days*) (score: 4). A summed CES-D score was used in the present study, and a higher CES-D score indicated a higher level of depression [26]. The CES-D has been found to be a valid and psychometrically sound instrument, including its Taiwan version [27,28]. The Cronbach’s α in the present study was 0.92.

#### 2.2.4. Anxiety

Participant anxiety was measured using the State-Trait Anxiety Inventory (STAI). The STAI used in the present study contains 20 items with responses ranging from *almost never* (score: 1) to *almost always* (score: 4). A summed STAI score was used in the present study, and a higher STAI score indicated a higher level of state anxiety [29]. The STAI has been found to be a valid and psychometrically sound instrument, including its Taiwan version [30,31,32,33]. The Cronbach’s α in the present study was 0.88.

#### 2.2.5. Loneliness

The loneliness of the participants was measured using the UCLA Loneliness Scale. The UCLA Loneliness Scale contains 20 items with responses ranging from *never* (score: 1) to *always* (score: 4). A summed UCLA Loneliness Scale score was used in the present study, and a higher score indicated a higher level of loneliness [34]. The UCLA Loneliness Scale has been found to be a valid and psychometrically sound instrument, including its Taiwan version [35,36]. The Cronbach’s α in the present study was 0.90.

#### 2.2.6. Demographics

The assessed participant demographics included their age, educational level, and sexual and gender orientation.

### 2.3. Data Analysis

All the data analyses were performed using the *psych* package [37] or *lavaan* package [38] of R software. The participants were split into two subsamples to avoid using the same sample for exploratory factor analysis (EFA) and confirmatory factor analysis (CFA). The two subsamples were separated using the *randbetween* function in Microsoft Excel. One subsample was named the EFA subsample (for EFA and parallel analysis), and the other was the CFA subsample (for CFA).

The characteristics of the entire sample and each subsample were analyzed using descriptive statistics, including mean (SD) and frequency (percentage). The item properties of the EASSSiHS were analyzed using frequency and percentage for the entire sample. Parallel analysis was used to determine the number of factors for the EASSSiHS and applied to 1000 Monte Carlo simulated samples and the EFA subsample. In the parallel analysis, if the actual eigenvalue of a factor calculated from the EFA subsample was higher than the 95% upper limit of the eigenvalue calculated from the simulated samples, the factor was considered to exist [39]. After the number of factors was determined, EFA examined the item–factor relationship using principal axis functioning extraction. In the EFA, an oblique rotation was used (if the factor number was two or more) with the Oblimin function. After the factor structure was identified using the parallel analysis and EFA results, the identified factor structure of EASSSiHS was further examined using CFA. For CFA, a maximum likelihood estimator was used. The fit indices that were adopted to examine whether CFA supported the identified factor structure were as follows: a nonsignificant χ^2^, comparative fit index (CFI) > 0.9, Tucker–Lewis index (TLI) > 0.9, root mean square error of approximation (RMSEA) < 0.08, and standardized root mean square residual (SRMR) < 0.08 [40].

Following the confirmation of the factor structure of EASSSiHS, the internal consistency of the EASSSiHS was examined using McDonald’s ω. It produced a value of >0.7, indicating good internal consistency [41]. Finally, the concurrent validity of the EASSSiHS was examined using Pearson correlations with perceived sexual stigma from family and friends, depression, anxiety, and loneliness.

## 3. Results

Table 1 displays participant characteristics, including the entire sample (*n* = 736), the subsample for EFA (*n* = 330), and the subsample for CFA (*n* = 406). No significant differences were found in the characteristics between the two subsamples (*p* > 0.05). The entire sample’s mean age was 31.03 (SD = 6.59) years, and their educational level was relatively high (nearly 90% had completed undergraduate degrees). The majority of the sample were gay (*n* = 611; 83.0%). The CES-D, STAI, UCLA Loneliness Scale, and HRSS scores are displayed in Table 1.

The EASSSiHS item properties are presented in Table 2. In total, 241 (32.7%) participants reported experiences or worries. The most reported was “afraid of seeking health-care services because you worry that your homosexual or bisexual identity will be disclosed” (*n* = 151; 20.5%), followed by “avoid seeking health-care service because you worry that your homosexual or bisexual identity is disclosed” (*n* = 111; 15.1%).

Parallel analysis of 1000 Monte Carlo simulated samples indicated that the EASSSiHS contains two factors, although only one factor had an eigenvalue greater than 1 (Figure 1). EFA further indicated that items 1–4 in the EASSSiHS belong to the same factor (designated experienced stigma) and items 5 and 6 in the EASSSiHS belong to another factor (designated anticipated stigma). Although our study had a significant χ^2^ test (*p* value = 0.04), the two-factor structure suggested by EFA results was confirmed and supported by other fit indices in CFA: CFI = 0.99, TLI = 0.98, RMSEA (90% CI) = 0.050 (0.010, 0.085), and SRMR = 0.033. Table 3 displays the factor loadings of the EASSSiHS derived from EFA and CFA. Moreover, the two factors were significantly associated (r = 0.22 in EFA; =0.49 in CFA).

McDonald’s omega coefficient (ω, 0.85) indicated that the internal consistency of the EASSSiHS was good. Regarding the concurrent validity of the EASSSiHS (Table 4), its total score was significantly associated with the CES-D, STAI, UCLA, and HRSS scores (r = 0.208 to 0.241; *p* values < 0.001). The anticipated stigma factor had stronger associations with CES-D, STAI, UCLA, and HRSS, when compared with the experienced stigma factor (r = 0.201 to 0.242 vs. r = 0.113 to 0.178; *p* values < 0.01).

In total, 127 (17.3%) participants reported that they had experienced sexual stigma, 168 (22.8%) participants reported that they anticipated sexual stigma in health-care services, and 114 (15.5%) participants reported that they anticipated but did not experience sexual stigma.

## 4. Discussion

Our EFA results indicated that the factor structure of the EASSSiHS included two factors: experienced stigma and anticipated stigma. The results of CFA further confirmed the two-factor structure. The EASSSiHS had good internal consistency and acceptable concurrent validity. The associations between sexual stigma on the EASSSiHS with perceived sexual stigma from family and friends, depression, anxiety, and loneliness varied between the two factors of the EASSSiHS.

The six items in the EASSSiHS were similar to those assessing sexual stigma in health-care workers among MSM and female sex workers in Sub-Saharan Africa [17]. The similarity indicates that sexual stigma is a health problem commonly seen in health-care services across countries and regions. Nearly one-third (32.7%) of participants in the present study reported having experienced or anticipated sexual stigma in health-care services, as measured by the items on the EASSSiHS; 17.3% and 22.8% of participants reported experiencing and anticipating sexual stigma, respectively. The rate of participants with experienced sexual stigma in health-care services in this study was similar to that of a previous survey in the United States in which 10% and 8% of LGB individuals reported experiencing harsh or abusive language and having been refused care due to their sexual orientation identity in healthcare settings, respectively [42]. The results indicated that experiencing or anticipating sexual stigma in conjunction with health-care services was not rare among gay and bisexual men and that health-care workers’ cultural competency should be increased to provide unprejudiced care for LGB individuals.

The present study found that more participants anticipated sexual stigma than experienced sexual stigma (22.8% vs. 17.3%). Notably, 15.5% of participants reported any anticipated but no experienced sexual stigma. However, both experienced and anticipated sexual stigma in health-care services were significantly associated with perceived sexual stigma from family members and friends. The result indicated that gay and bisexual men might generate their perceived sexual stigma in health-care services due to close friends and family or the public even though they have not personally experienced sexual stigma in health-care services. Interventions for improving sexual stigma in health-care services are required to reduce sexual stigma in the public and among the family and friends of LGB individuals.

The present study found that both experienced and anticipated sexual stigma in health-care services were significantly associated with depression, anxiety, and loneliness among gay and bisexual men. According to minority stress theory [8], experienced and anticipated sexual stigma in health-care services may compromise LGB individuals’ mental health directly or indirectly by changing their cognition, coping, emotional regulation, and social interaction. Alternatively, gay and bisexual men with depression, anxiety, or loneliness may have more chances to seek medical assistance and an increased risk of experiencing sexual stigmatization by health-care workers. The cross-sectional study design limited the possibility of determining the temporal relationship between sexual stigma in health-care services and mental health problems; therefore, sexual stigma in health-care services warrants active interventions to ensure mental health among LGB individuals. Our results indicate that anticipated sexual stigma had stronger associations with depression, anxiety, and loneliness than experienced sexual stigma. The results indicated that various dimensions of sexual stigma in health-care services had different associations with mental health problems among gay and bisexual men.

The findings of the present study highlight the value of developing strategies for the prevention of sexual stigma in health-care services. Intervention should be implemented in multiple aspects [42]. First, health care institutions and providers should establish nondiscrimination policies to prohibit bias and discrimination based on sexual orientation and provide culturally competent care to LGB individuals. Health care institutions and providers should also require health profession students and health professionals to undergo significant cultural competency training about sexual orientation to provide respectful and nondiscriminatory care to LGB individuals. Second, laws and anti-discrimination policies at the national level should require all providers to deliver to LGB individuals the same level of high-quality care afforded to others, as well as to develop and implement goals, policies, and plans to ensure that LGB individuals are treated fairly. Third, individuals and organizations should educate themselves and each other about LGB rights, and, when possible, educate health care providers about the needs of LGB individuals. Fourth, sexual stigma in health-care services requires further study, especially in the societies that traditionally have a lower tolerance of LGB individuals. The self-reported EASSSiHS developed in this study can be used to measure and compare the experienced and anticipated sexual stigma in health-care services among gay and bisexual men in various societies. This study examined the psychometric propensities of the EASSSiHS in only gay and bisexual men. The psychometric properties of the EASSSiHS for assessing experienced and anticipated sexual stigma in health-care services among lesbian and bisexual women warrant further study.

### Limitations

Our study has several limitations. First, our sample might have selection biases as we recruited participants by posting an online advertisement. Second, most of the participants were in young adulthood and had a high educational level. Whether the results of our study can be generalized to middle-aged, elderly, or low-educated gay and bisexual men warrants examination. Third, the cross-sectional study design limited our ability to determine the temporal relationship between sexual stigma in health-care services and mental health problems. Fourth, all data were collected from the participants’ self-report, and single-rater and recall biases cannot be fully controlled.

## 5. Conclusions

The results of our study support the psychometric properties of the newly developed EASSSiHS for assessing experienced and anticipated sexual stigma in health-care services in a sample of gay and bisexual men in Taiwan. Our study also revealed that the experiences and worries of sexual stigma in health-care services were not rare among gay and bisexual men; therefore, interventions for enhancing the cultural competency of health-care workers are urgently required.

## Figures and Tables

**Figure 1 ijerph-19-13877-f001:**
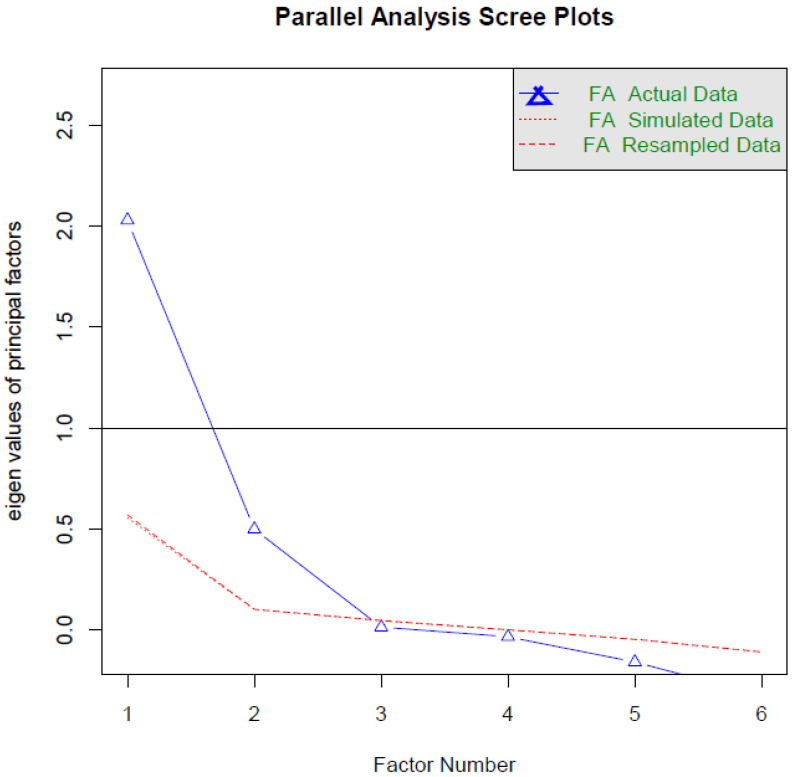
Parallel analysis of the Experienced and Anticipated Sexual Stigma Scale in Health-care Services. Two factors are recommended with the principal axis functioning extraction using 1000 simulated samples.

**Table 1 ijerph-19-13877-t001:** Participant characteristics.

	Entire Sample (*n* = 736)	EFA Sample (*n* = 330)	CFA Sample (*n* = 406)
Age (year), mean (SD)	31.03 (6.59)	31.50 (6.79)	30.65 (6.39)
Educational level, *n* (%)			
High school	79 (10.7)	30 (9.1)	49 (12.0)
Undergraduate	520 (70.6)	238 (72.1)	282 (69.5)
Postgraduate	137 (18.7)	62 (18.8)	75 (18.5)
Transgender, *n* (%)			
No	733 (99.6)	328 (99.4)	405 (99.8)
Yes	3 (0.4)	2 (0.6)	1 (0.2)
Sexual orientation, *n* (%)			
Gay	611 (83.0)	281 (85.2)	330 (81.3)
Bisexual	125 (17.0)	49 (14.8)	76 (18.7)
CES-D score, mean (SD)	18.17 (11.14)	17.93 (11.27)	18.37 (11.04)
STAI score, mean (SD)	39.58 (12.31)	39.40 (12.30)	39.72 (12.33)
UCLA score, mean (SD)	52.84 (4.96)	52.76 (5.04)	52.91 (4.89)
HRSS score, mean (SD)	26.72 (6.94)	26.50 (6.87)	26.90 (7.01)

CES-D = Center for Epidemiologic Studies Depression Scale; STAI = State-Trait Anxiety Inventory; UCLA = UCLA Loneliness Scale; HRSS = homosexuality-related stigma scale.

**Table 2 ijerph-19-13877-t002:** Item responses for the Experienced and Anticipated Sexual Stigma Scale in Health-care Services (*n* = 736).

Item Description	*n* (%)	
	Never Experienced	Ever Experienced
1. Rejection: Have been rejected from receiving health-care services because of your gay or bisexual identity	709 (96.3)	27 (3.7)
2. Inferior: Have received inferior health-care services because of your gay or bisexual identity	686 (93.2)	50 (6.8)
3. Gossiping: Have heard health-care service staff gossiping about your gay or bisexual identity	648 (88.0)	88 (12.0)
4. Difficulty: Have experienced difficulty receiving health-care services because of your gay or bisexual identity	706 (95.9)	30 (4.1)
5. Worry: Have been afraid of seeking health-care services because you worry that your gay or bisexual identity will be disclosed	585 (79.5)	151 (20.5)
6. Avoidance: Have avoided seeking health-care services because you worried that your gay or bisexual identity would be disclosed	625 (84.9)	111 (15.1)

**Table 3 ijerph-19-13877-t003:** Factor loadings of the Experienced and Anticipated Sexual Stigma Scale in Health-care Services.

	Exploratory Factor Analysis (*n* = 330)		Confirmatory Factor Analysis (*n* = 406)	
Item	Factor 1	Factor 2	Factor 1	Factor 2
1. Rejection	0.56	−0.10	0.56	
2. Inferiority	0.68	−0.08	0.64	
3. Gossiping	0.28	−0.07	0.48	
4. Difficulty	0.68	−0.24	0.80	
5. Worry	0.14	−0.86		0.85
6. Avoidance	0.22	−0.80		0.77

Note. Fit indices for confirmatory factor analysis: χ^2^ (df) = 16.14 (8), *p* value of χ^2^ = 0.04, comparative fit index = 0.99, Tucker–Lewis index = 0.98, root mean square error of approximation (90% CI) = 0.050 (0.010, 0.085), standardized root mean square residual = 0.033.

**Table 4 ijerph-19-13877-t004:** Concurrent validity of the Experienced and Anticipated Sexual Stigma Scale in Health-care Services (EASSSiHS).

		r (*p*)	
	EASSSiHS Total Score	Experienced Stigma Factor	Anticipated Stigma Factor
CES-D	0.241 (<0.001)	0.145 (<0.001)	0.242 (<0.001)
STAI	0.212 (<0.001)	0.113 (0.002)	0.227 (<0.001)
UCLA	0.208 (<0.001)	0.132 (<0.001)	0.201 (<0.001)
HRSS	0.237 (<0.001)	0.178 (<0.001)	0.202 (<0.001)

CES-D = Center for Epidemiologic Studies Depression Scale; STAI = State-Trait Anxiety Inventory; UCLA = UCLA Loneliness Scale; HRSS = homosexuality-related stigma scale.

## Data Availability

The data used in this study are available upon reasonable request to the corresponding authors.

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
