# Peer review of "A Newly Developed Scale for Assessing Experienced and Anticipated Sexual Stigma in Health-Care Services for Gay and Bisexual Men"

_ijerph, 2022, doi:10.3390/ijerph192113877_

Round 1
Reviewer 1 Report
This is an excellent paper that covers an important topic related to healthcare provision for LGB patients in Taiwan. I recommend publication with a few suggested changes as below.
Introduction:
- Please explain why trans and gender fluid patients were not included in the study
- Please include why women were not included in the study
- In the first paragraph, consider specifying a few of the physical diseases that are more prevelant among LGB patients
- Can you explain more what you mean by "evaluating multiple sources of sexual stigma simultaneously may increase clients' bias in reporting the experiences of sexual stigman from health-care providers"?
Materials and methods:
- Why were the three counseling centers chosen? Who did the online recruitment?
- Can you include more information about how the original EASSSiHS was developed? Was it developed as a result of the focus groups? If so, how were the original discussion topics and questions for the focus groups determined?
Results:
- Can you explain in table 1 what you mean when you include that 733 of the patients identified as transgender?
Discussion:
- Can you comment on whether the rates of experienced and anticipated stigma among the patient population in your study is more, less, or similar to rates experienced in other populations?
- Can you comment on any interventions either in Taiwan or in other countries that seek to address and reduce anticipated and experienced stigma among LGB patients? Do you have any specific suggestions for how this can be addressed in Taiwan specifically?
Author Response
We appreciated your valuable comments. As discussed below, we have revised our manuscript with underlines based on your suggestions. Please let us know if we need to provide anything else regarding this revision.
Comment 1
Introduction:
- Please explain why trans and gender fluid patients were not included in the study
Response
Thank you for your comment. This study did not exclude transgender or gender fluid individuals. Any man who identified as gay or bisexual, aged 20 or older, and living in Taiwan could participate in this study. There were three transgender participants in this study. Please refer to Table 1.
Comment 2
- Please include why women were not included in the study
Response
Thank you for your comment. We added the explanation for only including gay and bisexual men in this study into Introduction section as below. We also added the necessity of examining the psychometric propensities of the EASSSiHS in lesbian and bisexual women into Discussion section as below.
“…there were gender differences in the experienced and anticipated sexual stigma between gay and bisexual men and lesbian and bisexual women [22].” Please refer to line 89-90.
“This study examined the psychometric propensities of the EASSSiHS in only gay and bisexual men. The psychometric properties of the EASSSiHS for assessing experienced and anticipated sexual stigma in health-care services among lesbian and bisexual women warrant further study.” Please refer to line 316-319.
Comment 3
- In the first paragraph, consider specifying a few of the physical diseases that are more prevelant among LGB patients
Response
Thank you for your suggestion. We specified a few of the physical diseases that are more prevalent among LGB individuals as below in Introduction section. Please refer to line 44-46.
“Studies have indicated elevated rates of several physical diseases such as human immunodeficiency virus (HIV) infection [2], cancer [3], diabetes [4] and hypertension [5] among LGB individuals.”
Comment 4
- Can you explain more what you mean by "evaluating multiple sources of sexual stigma simultaneously may increase clients' bias in reporting the experiences of sexual stigma from health-care providers"?
Response
Thank you for your comment. We added explanations in Introduction section as below to make its meaning clearer. Please refer to line 70-73.
“Evaluating multiple sources of sexual stigma simultaneously may increase clients’ bias in identifying the sources of sexual stigma. Evaluating multiple sources of sexual stigma simultaneously may also reduce clients’ incentive to participate in the survey because of increased cognitive and emotional loading.”
Comment 5
Materials and methods:
- Why were the three counseling centers chosen? Who did the online recruitment?
Response
Thank you for your comment. We added explanations for choosing the three counseling centers into Methods section as below. The research team did the online recruitment. Please refer to line 101-104.
“We also posted advertisement on the home pages of three health promotion and counseling centers for LGB individuals. The three centers were organized by gay and lesbian friendly groups; advertisement on their home pages could gain attention and trust from gay and bisexual men.”
Comment 6
- Can you include more information about how the original EASSSiHS was developed? Was it developed as a result of the focus groups? If so, how were the original discussion topics and questions for the focus groups determined?
Response
We developed the EASSSiHS based on the results of the focus groups. We determined the discussion topics for the focus groups according to the results of literature review and the aims of this study. We described the process of conducting focus groups and determining the discussion topics in Methods section as below. Please refer to line 117-133.
“Before beginning the formal research, we conducted three focus group interviews from March to May 2021 to help develop the EASSSiHS assessing the experienced and anticipated sexual stigma when receiving health-care services among gay and bisexual men. … According to the results of literature review and the aims of this study, we determined the discussion topics for the focus groups as the experiences and worries of receiving treatment that differed from their expectations and different from that received by others due to the health-care providers’ awareness or suspicion of participants’ sexual orientation. The principal investigator led the group discussion and encouraged the members to express their opinions. Three researchers reviewed the transcript and coded the data for indications of sexual stigma experienced or anticipated by gay and bisexual men. The principal investigator reviewed the coding results and integrated them into six items of the EASSSiHS…”
Comment 7
Results:
- Can you explain in table 1 what you mean when you include that 733 of the patients identified as transgender?
Response
Thank you for your reminding. We apologized for the mistake. We corrected it into “3 transgender and 733 non-transgender participants” in Table 1.
Comment 8
Discussion:
- Can you comment on whether the rates of experienced and anticipated stigma among the patient population in your study is more, less, or similar to rates experienced in other populations?
Response
Thank you for your comment. We added the discussion regarding the comparison of experienced stigma between the results of this study and the previous study in the United States as below. Please refer to line 265-273.
“17.3% and 22.8% of participants reported experiencing and anticipating sexual stigma, respectively. The rate of participants with the experienced sexual stigma in health-care services in this study was similar to that of a previous survey in the United States in which 10% and 8% of LGB individuals reported experiencing harsh or abusive language and having been refused care due to their sexual orientation identity in healthcare settings, respectively [42]. The results indicated that experiencing or anticipating sexual stigma in conjunction with health-care services was not rare among gay and bisexual men and that health-care workers’ cultural competency should be increased to provide unprejudiced care for LGB individuals.”
Comment 9
- Can you comment on any interventions either in Taiwan or in other countries that seek to address and reduce anticipated and experienced stigma among LGB patients? Do you have any specific suggestions for how this can be addressed in Taiwan specifically?
Response
Thank you for your comment. We added the suggestions for improving sexual stigma in health-care services from four aspects as below. Please refer to line 299-319.
“The findings of the present study highlight the value of developing strategies for the prevention of sexual stigma in health-care services. Intervention should be implemented in multiple aspects [42]. First, health care institutions and providers should establish nondiscrimination policies to prohibit bias and discrimination based on sexual orientation and provide culturally competent care to LGB individuals. Health care institutions and providers should also require health profession students and health professionals to undergo significant cultural competency training about sexual orientation to provide respectful and nondiscriminatory care to LGB individuals. Second, laws and anti-discrimination policies at the national level should require all providers to deliver to LGB individuals the same level of high-quality care afforded others as well as to develop and implement goals, policies and plans to ensure that LGB individuals are treated fairly. Third, individuals and organizations should educate themselves and each other about LGB rights, and when possible, educate health care providers about the needs of LGB individuals. Fourth, sexual stigma in health-care services requires further study, especially in the societies that traditionally have a lower tolerance of LGB individuals. The self-reported EASSSiHS developed in this study can be used to measure and compare the experienced and anticipated sexual stigma in health-care services among gay and bisexual men in various societies. This study examined the psychometric propensities of the EASSSiHS in only gay and bisexual men. The psychometric properties of the EASSSiHS for assessing experienced and anticipated sexual stigma in health-care services among lesbian and bisexual women warrant further study.”
Reviewer 2 Report
In this paper, the authors adopted a mix-methods approach to develop an assessing scale for sexual stigma in healthcare services for gay and bisexual men in Taiwan (although mainly quant data is reported here). The methodological approach followed from the initial qual data collection arm from FGDs with 24 participants (briefly reported in this paper) and the parallel quant data collection from a cross-sectional convenience sample of 736 men recruited from a range of online community resources. Both exploratory and confirmatory factor analyses (Monte Carlo simulation applied in the CFA) were used by splitting the entire survey sample. The final two-factor structure seems meaningful and parsimonious.
My suggestion would be in the conclusion section:
1) briefly discuss the impact of COVID during the data collection period
2) future implementation and further adaptation of this newly developed scale in a broad GBM samples in the Asia Pacific Region, and perhaps in the context of self-reported quality of life improvement.
Author Response
We appreciated your valuable comments. As discussed below, we have revised our manuscript with underlines based on your suggestions. Please let us know if we need to provide anything else regarding this revision.
Comment 1
Briefly discuss the impact of COVID during the data collection period
Response
Taiwan experienced two severe COVID-19 outbreaks. The first outbreak occurred between May and July 2021; the second occurred since May 2022 to now. This study was conducted between August 2021 and May 2022 and thus was not impacted by the COVID-19. We added the explanation into Methods section as below. Please refer to line 99-101.
“…from August 2021 to May 2022, starting since the end of the first severe COVID-19 outbreak (between May and July 2021) and ending at the beginning of the second severe outbreak (since May 2022 to now).”
Comment 2
Future implementation and further adaptation of this newly developed scale in a broad GBM samples in the Asia Pacific Region, and perhaps in the context of self-reported quality of life improvement.
Response
Thank you for your comment. We added the suggestions for improving sexual stigma in health-care services and further application of the EASSSiHS developed in this study as below. Please refer to line 299-319.
“The findings of the present study highlight the value of developing strategies for the prevention of sexual stigma in health-care services. Intervention should be implemented in multiple aspects [42]. First, health care institutions and providers should establish nondiscrimination policies to prohibit bias and discrimination based on sexual orientation and provide culturally competent care to LGB individuals. Health care institutions and providers should also require health profession students and health professionals to undergo significant cultural competency training about sexual orientation to provide respectful and nondiscriminatory care to LGB individuals. Second, laws and anti-discrimination policies at the national level should require all providers to deliver to LGB individuals the same level of high-quality care afforded others as well as to develop and implement goals, policies and plans to ensure that LGB individuals are treated fairly. Third, individuals and organizations should educate themselves and each other about LGB rights, and when possible, educate health care providers about the needs of LGB individuals. Fourth, sexual stigma in health-care services requires further study, especially in the societies that traditionally have a lower tolerance of LGB individuals. The self-reported EASSSiHS developed in this study can be used to measure and compare the experienced and anticipated sexual stigma in health-care services among gay and bisexual men in various societies. This study examined the psychometric propensities of the EASSSiHS in only gay and bisexual men. The psychometric properties of the EASSSiHS for assessing experienced and anticipated sexual stigma in health-care services among lesbian and bisexual women warrant further study.”